# Sequential Indeterminate Probability Theory for Multivariate Time Series Forecasting

## Abstract

Currently, there is no mathematical analytical form for a general posterior, however, Indeterminate Probability Theory Anonymous (2024b) has now discovered a way to address this issue. This is a big discovery in the field of probability and it is also applicable to multivariate time series (MTS) forecasting. Deep models, particularly transformer-based models, have shown better performance for MTS forecasting than traditional statistical models, however, deep models are black-boxes for human. In this paper, we propose a new probabilistic method for MTS forecasting that does not rely on any neural models, and this method does not require any training process. We formulate MTS forecasting problem as a complex posterior and consider the MTS value as an indeterminate probability distribution. Based on the indeterminate probability theory, the posterior becomes analytical tractable, even in the presence of a thousand-dimensional latent space. Experimental results show that our method outperforms LSTM models as well as some transformer-based models.

## 1 Introduction

Time series forecasting is a long-standing task that has a wide range of applications, including, weather forecasting, financial investment, Traffic estimation, etc. Zeng et al. (2022) With the development of deep learning, many models have been proposed and achieves very good performance for MTS forecasting Vaswani et al. (2017); Zhou et al. (2021); Wu et al. (2021); Zhou et al. (2022); Zhang & Yan (2023). However, deep models are black-boxes to human Buhrmester et al. (2019), as we do not know the mechanisms behind the model predictions.

In our opinion, the MTS forecasting problem can be formulated very well as a posterior. That is, given the past, we can infer the future, just like the meaning of a posterior. However, such a formulation has not been used in past works because the complex posterior does not have an analytical form and is not tractable. Now, with Indeterminate Probability Theory we can solve the MTS forecasting problem with a complex posterior and this method is not black-box anymore.

The rest of this paper is organized as follows: In Sec. 2, related works of MTS forecasting methods and Indeterminate Probability Theory are introduced. In Sec. 3, we use a simple time series example to explain the core idea of our proposed method, as well as the limitations of our method. In Sec. 4, the MTS forecasting problem is formulated as a complex posterior, and MTS values are considered as indeterminate probability distribution. In Sec. 5, it shows that our method outperforms LSTM, LSTnet, Transformer and Informer in three datasets, and abuse test is designed for the robustness checking of our method. Finally, we conclude the paper in Sec. 6.

## 2 Related Work

### 2.1 MTS Forecasting

**MTS forecasting** methods can be roughly divided into statistical methods and deep models. ARIMA Ariyo et al. (2014), SVAR Kilian & Lütkepohl (2017) and VARMA Scherrer & Deistler (2019) are typical statistical models, which assume linear cross-dimension and cross-time dependency Zhang & Yan (2023). Generally speaking, these statistical models require significant domain

expertise to build, and their performances are sometimes unsatisfactory. Therefore, deep models have been well developed for the MTS forecasting tasks.

**Transformers in Forecasting** A lot of work has been done to design new Transformer variants for MTS forecasting in recent years. Informers Zhou et al. (2021) proposes a ProbSparse self-attention mechanism, distilling techniques and generative style decoder to solve the long sequence time series forecasting problem. LogTrans Li et al. (2019) uses convolution self-attention with LogSparse design to utilize local information and reduce the complexity. FEDformer Zhou et al. (2022) uses Fourier enhanced method to get a linear complexity. Nie et al. (2023) The Transformer-based models for MTS forecasting are the most popular methods. However, these models are black-boxes to human, as we do not know the mechanisms behind the model predictions. In contrast, our proposed method does not use any neural network and is not black-box.

**Cross Time and Cross Dimension Dependencies** are critical for MTS forecasting, because information from associated series in other dimensions may improve prediction Zhang & Yan (2023). MICN Wang et al. (2023) propose a new framework for modelling local and global correlations of time series along with a new module instead of attention mechanism. WinIT Leung et al. (2023) propose Windowed Feature Importance in Time to address the temporal dependencies. PatchTST Nie et al. (2023) is also designed for local semantic information and attending longer history. DLinear model Zeng et al. (2022) argues that the nature of the permutation-invariant self-attention mechanism will result in temporal information loss. The work Wen et al. (2023) summarizes that spatio-temporal forecasting needs to take into account both temporal and spatio-temporal dependencies. There are more works researched that cross time and cross dimension dependencies are very important factors for MTS forecasting. This may be an important reason why the performances of traditional statistical methods are not better than deep models, because they cannot utilize these dependencies. In this paper, our proposed method is a complex posterior which can use both cross time and cross dimension dependencies, even for very long-term time series forecasting, the posterior can effectively leverage these dependencies. For more details, please refer to Sec. 3.

## 2.2 INDETERMINATE PROBABILITY THEORY

Special random variable $X \in \{x_1, x_2, \ldots, x_n\}$ is defined for random experiments, and $X = x_k$ is for $k^{th}$ experiment, so $P(x_k) \equiv 1$. Random variable $Y \in \{y_1, y_2, \ldots, y_m\}$ is a general discrete variable (continuous variable is also allowed), $P(y_l|x_k) = y_l(k) \in [0,1]$ is the indeterminate probability to describe the observed outcome of sample $x_k$. $P^{\mathbf{z}}(y_l \mid x_t)$ is for the inference outcome of sample $x_t$, superscript $\mathbf{z}$ stands for the medium − N-dimensional latent random variables $\mathbf{z} = \left(z^1, z^2, \ldots, z^N\right)$, via which we can infer $Y = y_l, l = 1, 2, \ldots, m$.

The analytical inference probability with the posterior is Anonymous (2024b)

$$P^{\mathbf{z}}(y_l \mid x_t) = \int_{\mathbf{z}} \left( \frac{\sum_{k=1}^n \left(P\left(y_l \mid x_k\right) \cdot P\left(\mathbf{z} \mid x_k\right)\right)}{\sum_{k=1}^n P\left(\mathbf{z} \mid x_k\right)} \cdot P\left(\mathbf{z} \mid x_t\right) \right), \tag{1}$$

where

$$P\left(\mathbf{z} \mid x_k\right) = P\left(z^1, z^2, \ldots, z^N \mid x_k\right) = \prod_{i=1}^{N} P\left(z^i \mid x_k\right), \tag{2}$$

and $P\left(\mathbf{z} \mid x_t\right)$ is similar.

## 3 BACKGROUND

We learn from the past.

Why can we learn from the past? Because the past is similar to the present. This is the core idea of our method.

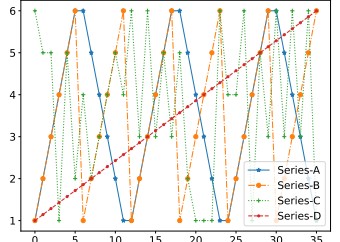

Table 1: Next point inference probability given 0,1,2 past outcomes.

|         | 0-Given | 1-Given | 2-Given |
|---------|---------|---------|---------|
| Series-A | 1/6 | 1/2 | 1 |
| Series-B | 1/6 | 1 | 1 |
| Series-C | 1/6 | 1/6 | 1/6 |
| Series-D | - | - | - |

Figure 1: Example of Time Series.

Let's first see a time series example in Figure 1. For series-A, if we do not have any past information, the probability of the next outcome is 1/6. If we know one past outcome, the probability of the next outcome becomes 1/2. If we have more information, we can make a better inference. Details can be found in Table 1.

Besides, for series-A, if we know one past outcome of series-B without knowing the past outcome of series-A, the probability of the next outcome of series-A becomes 1/2. This means that for a probabilistic model, the information from other series (random variable) is also useful.

In other words, a probabilistic model can utilize both cross-time and cross-dimension information, Crossformer Zhang & Yan (2023) also shows such a functionality.

However, for series-D, a probabilistic model is not able to make good predictions because the past differs from the present. This is a limitation of our proposed method. As shown in Table 2, our results with WTH dataset are significantly worse than those of other methods, because the WTH dataset exhibits a similar trend to series-D.

In addition, how can we judge if the past is similar to the present mathematically?

We introduce a Gaussian variance factor as a critical hyperparameter in our method to determine whether the past is similar to the present and compatible with noise from dataset. The intuitive explanation is shown in Figure 2.

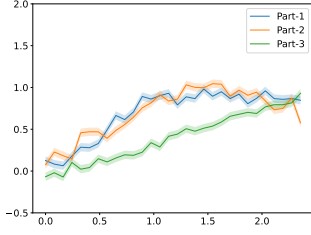
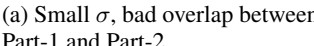
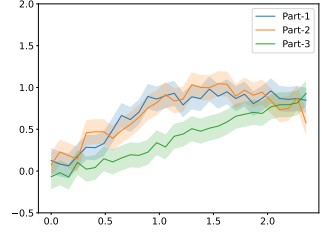
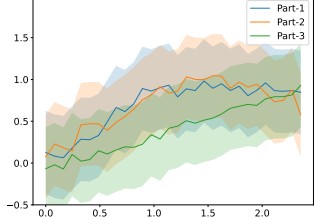

(a) Small $\sigma$, bad overlap between Part-1 and Part-2.

(b) Proper $\sigma$, good overlap between Part-1 and Part-2.

(c) Large $\sigma$, bad overlap between Part-1 and Part-3.

Figure 2: Effect of Gaussian variance $\sigma$. For our proposed method, the series are not in a 2-D space, the overlap happens in a high latent space.

# 4 SEQUENTIAL INDETERMINATE PROBABILITY THEORY

## 4.1 POINT VALUE INTERPRETATION

For a point value, there are two directions of interpretations that can be considered as indeterminate probability distributions: the probability value or the parameter of the probability distribution.

And both interpretations are used in our proposed method.

For MTS value, we first transform the value to $[0, 1]$, and let the point value be $f^i(t) \in [0, 1], t \in \mathbb{Z}$ for time series $i = 1, 2, \ldots, N$.

In Appendix A, we have mathematically and rigorously proved that the following interpretations can provide a strict and exact forecasting for a general periodic continuous function. Therefore, the sequential indeterminate probability in this paper is formulated as

$$P(z^i \mid x_t) = \mathcal{N}(z; f^i(t), \sigma^2), \qquad i = 1, 2, \ldots, N. \text{ Gaussian distribution} \qquad (3)$$

$$P(y_1^I \mid x_t) = f^I(t), \qquad I = 1, 2, \ldots, N. \text{ Bernoulli distribution} \qquad (4)$$

Where $\sigma$ is a critical hyperparameter of our method, an intuitive explanation of $\sigma$ see Figure 2. And $P(y_2^I \mid x_t) = 1 - f^I(t)$ does not need to be focused in this paper.

## 4.2 Formulation of Sequential Probabilistic Problem

Let $L$ denotes the length of past series, $D$ denotes the length of predicted series, $N$ denotes the number of series.

According to the definition of indeterminate probability, we have

$$P\left(z^{i,-j} \mid x_k\right) = P\left(z^i \mid x_{k-j}\right), \qquad j = 0, 1, \ldots, L. \qquad (5)$$

$$P\left(y_1^{I,+d} \mid x_k\right) = P\left(y_1^I \mid x_{k+d}\right), \qquad d = 1, 2, \ldots, D. \qquad (6)$$

where $i = 1, 2, \ldots, N$ and $I = 1, 2, \ldots, N$.

Let

$$\begin{aligned}
\mathbf{z}^{:-L} &:= \left(\mathbf{z}^0, \mathbf{z}^{-1}, \ldots, \mathbf{z}^{-L}\right) \\
&= \left(z^{1,0}, z^{2,0}, \ldots, z^{N,0}; z^{1,-1}, z^{2,-1}, \ldots, z^{N,-1}; \ldots; z^{1,-L}, z^{2,-L}, \ldots, z^{N,-L}\right)
\end{aligned} \qquad (7)$$

and

$$\begin{aligned}
\mathbf{Y}^{:+D} &:= \left(\mathbf{Y}^{+1} = \mathbf{y_1}^{+1}, \mathbf{Y}^{+2} = \mathbf{y_1}^{+2}, \ldots, \mathbf{Y}^{+D} = \mathbf{y_1}^{+D}\right) \\
&= \left(y_1^{1,+1}, y_1^{2,+1}, \ldots, y_1^{N,+1}; y_1^{1,+2}, y_1^{2,+2}, \ldots, y_1^{N,+2}; \ldots; y_1^{1,+D}, y_1^{2,+D}, \ldots, y_1^{N,+D}\right)
\end{aligned} \qquad (8)$$

According to the Candidate Axioms that given $X$, all $z^{i,-j}$ are conditional independent Anonymous (2024b), therefore, with Eq. (5) we have

$$P\left(\mathbf{z}^{:-L} \mid x_k\right) = \prod_{j=0}^{L} \prod_{i=1}^{N} P\left(z^{i,-j} \mid x_k\right) = \prod_{j=0}^{L} \prod_{i=1}^{N} P\left(z^i \mid x_{k-j}\right) \qquad (9)$$

The problem of MTS forecasting is then formulated as the following inference probability, and according to the Hypothesis that given $X$, all the inference probability $P^{\mathbf{z}^{:-L}}\left(y_1^{i,+d} \mid x_t\right)$ are also conditional independent Anonymous (2024b), we have

$$P^{\mathbf{z}^{:-L}}\left(\mathbf{Y}^{:+D} \mid x_t\right) = \prod_{d=1}^{D} \prod_{I=1}^{N} P^{\mathbf{z}^{:-L}}\left(y_1^{I,+d} \mid x_t\right) \qquad (10)$$

Substitute Eq. (9) into Eq. (1), which is similar to the derivation process in CIPNN Anonymous (2024a), together with Eq. (6) we have

$$P^{\mathbf{z}^{:-L}}\left(y_1^{I,+d} \mid x_t\right)$$

$$= \int\limits_{\mathbf{z}^{:-L}} \left( \frac{\sum_{k=L+1}^{n-D} \left(P\left(y_1^I \mid x_{k+d}\right) \cdot P\left(\mathbf{z}^{:-L} \mid x_k\right)\right)}{\sum_{k=L+1}^{n-D} P\left(\mathbf{z}^{:-L} \mid x_k\right)} \cdot P\left(\mathbf{z}^{:-L} \mid x_t\right) \right) \tag{11}$$

$$= \int\limits_{\mathbf{z}^{:-L}} \left( \frac{\sum_{k=L+1}^{n-D} \left(P\left(y_1^I \mid x_{k+d}\right) \cdot \prod_{j=0}^{L} \prod_{i=1}^{N} P\left(z^i \mid x_{k-j}\right)\right)}{\sum_{k=L+1}^{n-D} \prod_{j=0}^{L} \prod_{i=1}^{N} P\left(z^i \mid x_{k-j}\right)} \cdot \prod_{j=0}^{L} \prod_{i=1}^{N} P\left(z^i \mid x_{t-j}\right) \right) \tag{12}$$

$$= \mathbb{E}_{z^i \sim P(z^i \mid x_{t-j})} \left[ \frac{\sum_{k=L+1}^{n-D} \left(P\left(y_1^I \mid x_{k+d}\right) \cdot \prod_{j=0}^{L} \prod_{i=1}^{N} P\left(z^i \mid x_{k-j}\right)\right)}{\sum_{k=L+1}^{n-D} \prod_{j=0}^{L} \prod_{i=1}^{N} P\left(z^i \mid x_{k-j}\right)} \right] \tag{13}$$

Substitute Eq.(3) and Eq.(4) into Eq.(13)

$$P^{\mathbf{z}^{:-L}}(y_1^{I,+d} \mid x_t)$$

$$= \mathbb{E}_{z \sim \mathcal{N}(z; f^i(t-j), \sigma^2)} \left[ \frac{\sum_{k=L+1}^{n-D} \left(f^I(k+d) \cdot \prod_{j=0}^{L} \prod_{i=1}^{N} \mathcal{N}(z; f^i(k-j), \sigma^2)\right)}{\sum_{k=L+1}^{n-D} \prod_{j=0}^{L} \prod_{i=1}^{N} \mathcal{N}(z; f^i(k-j), \sigma^2)} \right] \tag{14}$$

In this way, we can get our predicted MTS values. The further implementation of this equation has already been discussed in CIPNN and CIPAE Anonymous (2024a), we will not further discuss it in this paper.

Finally, $P^{\mathbf{z}^{:-L}}\left(y_1^{I,+d} \mid x_t\right)$ is the predicted MTS value, and $P\left(y_1^{I,+d} \mid x_t\right) = P\left(y_1^I \mid x_{t+d}\right) = f^I(t+d)$ is the ground truth MTS value.

## 5 EXPERIMENTS AND RESULTS

Our proposed method does not need any training process, we only need to load the train data, and put the train data and test data together into Eq. (14), we can then get the predicted results.

### 5.1 PROTOCOLS

**Datasets**  We conduct experiments on six real-world datasets following Zhou et al. (2021); Zhang & Yan (2023): ETTh1, ETTm1, WTH, ECL and Traffic. The data split for all datasets are same to Crossformer Zhang & Yan (2023).

**Baselines**  We use the following baselines which are mainly same to Crossformer Zhang & Yan (2023): LSTMa Bahdanau et al. (2014), LSTnet Lai et al. (2017), MTGNN Wu et al. (2020), Transformer Vaswani et al. (2017), Informer Zhou et al. (2021), Autoformer Wu et al. (2021), FEDformer Zhou et al. (2022), Crossformer Zhang & Yan (2023).

**Setup**  Our setup is summarized in Figure 3. train/val/test sets are firstly normalized with StandardScaler using the mean and standard deviation of training set, which is the same as Zhou et al. (2021); Zhang & Yan (2023). We further transform these sets with MinMaxScaler using the scaler factor of the training set. Using Eq. (14), we obtain the predictions, and they need to be inversely transformed using the scaler factor of training set for the final evaluation. Our proposed method is not separable, so we do not have any ablation test.

### 5.2 MAIN RESULTS

As shown in Table 2, our proposed method outperforms LSTM, LSTnet, Transformer and Informer in three datasets, the detailed hyperparameter settings are listed in Table 4. Besides, we can see that

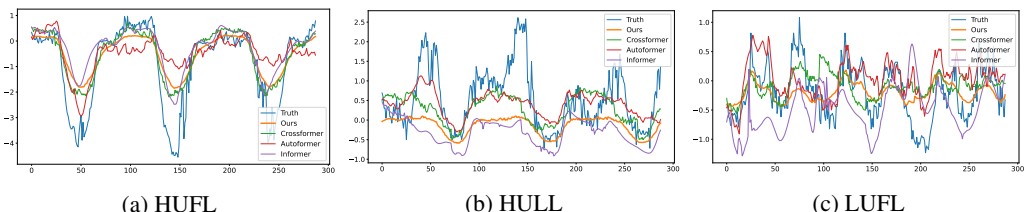

Figure 3: Process of MTS forecasting of our method.

our proposed method can achieve quite competitive results by using only 4 past points ($L + 1 = 4$) on ECL. Furthermore, the dimensionality of our latent space are quite large, up to 1724-D. This may be the power of analytical solution.

Results on WTH dataset are very bad, the reason has already been analyzed in Sec. 3. In our opinion, this limitation can be optimized with some tricks, such as making some conversions to give MTS a good periodic property. We will not further discuss it in this paper.

Table 2: Multivariate long-term forecasting errors in terms of MSE and MAE, the lower the better. Results with green color are for the methods not better than ours, the best results are highlighted in **bold**. Results of other methods are from Crossformer Zhang & Yan (2023). Tests are repeated for 3 times and the mean values are reported.

| Methods | | LSTMa | | LSTnet | | MTGNN | | Transformer | | Informer | | Autoformer | | FEDformer | | Crossformer | | *Ours\** |
| Metric | | MSE | MAE | MSE | MAE | MSE | MAE | MSE | MAE | MSE | MAE | MSE | MAE | MSE | MAE | MSE | MAE | MSE | MAE |
|---|---|---|---|---|---|---|---|---|---|---|---|---|---|---|---|---|---|---|---|
| ETTh1 | 24 | 0.650 | 0.624 | 1.293 | 0.901 | 0.336 | 0.393 | 0.620 | 0.577 | 0.577 | 0.549 | 0.439 | 0.440 | 0.318 | 0.384 | **0.305** | **0.367** | 0.666 | 0.609 |
| | 48 | 0.720 | 0.675 | 1.456 | 0.960 | 0.386 | 0.429 | 0.692 | 0.671 | 0.685 | 0.625 | 0.429 | 0.442 | **0.342** | 0.396 | 0.352 | **0.394** | 0.704 | 0.627 |
| | 168 | 1.212 | 0.867 | 1.997 | 1.214 | 0.466 | 0.474 | 0.947 | 0.797 | 0.931 | 0.752 | 0.493 | 0.479 | 0.412 | 0.449 | **0.410** | **0.441** | 0.804 | 0.680 |
| | 336 | 1.424 | 0.994 | 2.655 | 1.369 | 0.736 | 0.643 | 1.094 | 0.813 | 1.128 | 0.873 | 0.509 | 0.492 | 0.456 | 0.474 | **0.440** | **0.461** | 0.921 | 0.747 |
| | 720 | 1.960 | 1.322 | 2.143 | 1.380 | 0.916 | 0.750 | 1.241 | 0.917 | 1.215 | 0.896 | 0.539 | 0.537 | 0.521 | **0.515** | **0.519** | 0.524 | 0.935 | 0.766 |
| ETTm1 | 24 | 0.621 | 0.629 | 1.968 | 1.170 | 0.260 | 0.324 | 0.306 | 0.371 | 0.323 | 0.369 | 0.410 | 0.428 | 0.290 | 0.364 | **0.211** | **0.293** | 0.656 | 0.570 |
| | 48 | 1.392 | 0.939 | 1.999 | 1.215 | 0.386 | 0.408 | 0.465 | 0.470 | 0.494 | 0.503 | 0.485 | 0.464 | 0.342 | 0.396 | **0.300** | **0.352** | 0.791 | 0.644 |
| | 96 | 1.339 | 0.913 | 2.762 | 1.542 | 0.428 | 0.446 | 0.681 | 0.612 | 0.678 | 0.614 | 0.502 | 0.476 | 0.366 | 0.412 | **0.320** | **0.373** | 0.694 | 0.614 |
| | 288 | 1.740 | 1.124 | 1.257 | 2.076 | 0.469 | 0.488 | 1.162 | 0.879 | 1.056 | 0.786 | 0.604 | 0.522 | **0.398** | 0.433 | 0.404 | **0.427** | 0.766 | 0.656 |
| | 672 | 2.736 | 1.555 | 1.917 | 2.941 | 0.620 | 0.571 | 1.231 | 1.103 | 1.192 | 0.926 | 0.607 | 0.530 | **0.455** | **0.464** | 0.569 | 0.528 | 0.840 | 0.696 |
| WTH | 24 | 0.546 | 0.570 | 0.615 | 0.545 | 0.307 | 0.356 | 0.349 | 0.397 | 0.335 | 0.381 | 0.363 | 0.396 | 0.357 | 0.412 | **0.294** | **0.343** | 1.466 | 0.735 |
| | 48 | 0.829 | 0.677 | 0.660 | 0.589 | 0.388 | 0.422 | 0.386 | 0.433 | 0.395 | 0.459 | 0.456 | 0.462 | 0.428 | 0.458 | **0.370** | **0.411** | 2.020 | 0.874 |
| | 168 | 1.038 | 0.835 | 0.748 | 0.647 | 0.498 | 0.512 | 0.613 | 0.582 | 0.608 | 0.567 | 0.574 | 0.548 | 0.564 | 0.541 | **0.473** | **0.494** | 4.183 | 1.338 |
| | 336 | 1.657 | 1.059 | 0.782 | 0.683 | 0.506 | 0.523 | 0.707 | 0.634 | 0.702 | 0.620 | 0.600 | 0.571 | 0.533 | 0.536 | **0.495** | **0.515** | 5.340 | 1.536 |
| | 720 | 1.536 | 1.109 | 0.851 | 0.757 | **0.510** | **0.527** | 0.834 | 0.741 | 0.831 | 0.731 | 0.587 | 0.570 | 0.562 | 0.557 | 0.526 | 0.542 | 5.862 | 1.629 |
| ECL | 48 | 0.486 | 0.572 | 0.369 | 0.445 | 0.173 | 0.280 | 0.334 | 0.399 | 0.344 | 0.393 | 0.241 | 0.351 | 0.229 | 0.338 | **0.156** | **0.255** | 0.665 | 0.577 |
| | 168 | 0.574 | 0.602 | 0.394 | 0.476 | 0.236 | 0.320 | 0.353 | 0.420 | 0.368 | 0.424 | 0.299 | 0.387 | 0.263 | 0.361 | **0.231** | **0.309** | 0.692 | 0.589 |
| | 336 | 0.886 | 0.795 | 0.419 | 0.477 | 0.328 | 0.373 | 0.381 | 0.439 | 0.381 | 0.431 | 0.375 | 0.428 | **0.305** | 0.386 | 0.323 | **0.369** | 0.700 | 0.592 |
| | 720 | 1.676 | 1.095 | 0.556 | 0.565 | 0.422 | **0.410** | 0.391 | 0.438 | 0.406 | 0.443 | 0.377 | 0.434 | **0.372** | 0.434 | 0.404 | 0.423 | 0.712 | 0.596 |
| | 960 | 1.591 | 1.128 | 0.605 | 0.599 | 0.471 | 0.451 | 0.492 | 0.550 | 0.460 | 0.548 | **0.366** | **0.426** | 0.393 | 0.449 | 0.433 | 0.438 | 0.725 | 0.603 |
| ILI | 24 | 4.220 | 1.335 | 4.975 | 1.660 | 4.265 | 1.387 | 3.954 | 1.323 | 4.588 | 1.462 | 3.101 | 1.238 | **2.687** | **1.147** | 3.041 | 1.186 | 4.176 | 1.428 |
| | 36 | 4.771 | 1.427 | 5.322 | 1.659 | 4.777 | 1.496 | 4.167 | 1.360 | 4.845 | 1.496 | 3.397 | 1.270 | **2.887** | **1.160** | 3.406 | 1.232 | 4.055 | 1.394 |
| | 48 | 4.945 | 1.462 | 5.425 | 1.632 | 5.333 | 1.592 | 4.746 | 1.463 | 4.865 | 1.516 | 2.947 | 1.203 | **2.797** | **1.155** | 3.459 | 1.221 | 4.128 | 1.398 |
| | 60 | 5.176 | 1.504 | 5.477 | 1.675 | 5.070 | 1.552 | 5.219 | 1.553 | 5.212 | 1.576 | 3.019 | 1.202 | **2.809** | **1.163** | 3.640 | 1.305 | 4.358 | 1.433 |
| Traffic | 24 | 0.668 | 0.378 | 0.648 | 0.403 | 0.506 | 0.278 | 0.597 | 0.332 | 0.608 | 0.334 | 0.550 | 0.363 | 0.562 | 0.375 | **0.491** | **0.274** | 1.604 | 0.826 |
| | 48 | 0.709 | 0.400 | 0.709 | 0.425 | **0.512** | 0.298 | 0.658 | 0.369 | 0.644 | 0.359 | 0.595 | 0.376 | 0.567 | 0.374 | 0.519 | **0.295** | 1.658 | 0.845 |
| | 168 | 0.900 | 0.523 | 0.713 | 0.435 | 0.521 | 0.319 | 0.664 | 0.363 | 0.660 | 0.391 | 0.649 | 0.407 | 0.607 | 0.385 | **0.513** | **0.289** | 1.659 | 0.851 |
| | 336 | 1.067 | 0.599 | 0.741 | 0.451 | 0.540 | 0.335 | 0.654 | 0.358 | 0.747 | 0.405 | 0.624 | 0.388 | 0.624 | 0.389 | **0.530** | **0.300** | 1.708 | 0.861 |
| | 720 | 1.461 | 0.787 | 0.768 | 0.474 | **0.557** | 0.343 | 0.685 | 0.370 | 0.792 | 0.430 | 0.674 | 0.417 | 0.623 | 0.378 | 0.573 | **0.313** | 1.717 | 0.862 |

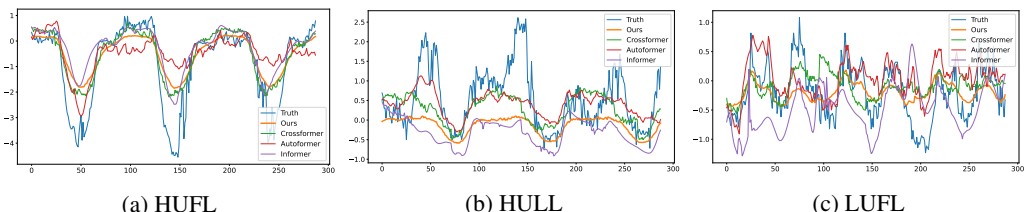

(a) HUFL      (b) HULL      (c) LUFL

Figure 4: Forecasting results of different methods on ETTm1 datasets.

## 5.3 ABUSE TEST

Indeterminate Probability Theory has a Candidate Axiom assumption: given $X$, all latent random variables are conditionally independent. We use an abuse test as critical evidence of this axiom. As shown in Table 3, even with duplicated random variables, our proposed method works just as well.

Table 3: Duplicated random variables for abuse test.

| Methods | | Original Dataset | | Duplicated Dataset | |
|---|---|---|---|---|---|
| Metric | | MSE | MAE | MSE | MAE |
| ETTh1 | 24 | 0.666 | 0.609 | 0.779 | 0.635 |
| | 48 | 0.704 | 0.627 | 1.137 | 0.762 |
| | 168 | 0.804 | 0.680 | 0.869 | 0.691 |
| | 336 | 0.921 | 0.747 | 0.959 | 0.743 |
| | 720 | 0.935 | 0.766 | 0.937 | 0.740 |
| ETTm1 | 24 | 0.656 | 0.570 | 0.690 | 0.576 |
| | 48 | 0.791 | 0.644 | 0.878 | 0.660 |
| | 96 | 0.694 | 0.614 | 0.704 | 0.609 |
| | 288 | 0.766 | 0.656 | 0.779 | 0.651 |
| | 672 | 0.840 | 0.696 | 0.851 | 0.687 |

## 5.4 HYPERPARAMETER ANALYSIS

The most critical hyperparameter for our proposed method is the Gaussian variance $\sigma$.

Similar to the analysis in Figure 2, for the same setup, our method fails to make predictions with very small $\sigma = 0.1$, because we cannot have a good overlap with past series. On the other hand, the predictions becomes very smooth with a large $\sigma = 2$ due to the unnecessary overlap, see Figure 5.

In addition, according to this analysis that we use a bigger $\sigma$ for large latent space and a smaller $\sigma$ for small latent space, as shown in Table 4.

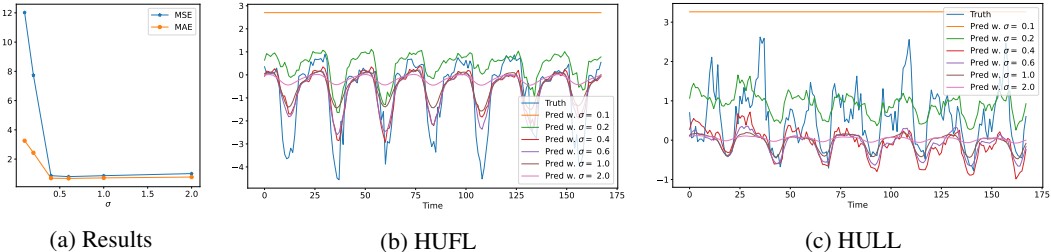

| (a) Results | (b) HUFL | (c) HULL |

Figure 5: Impact analysis of hyperparameter $\sigma$ on ETTh1, $\sigma = 0.6$ is the best.

## 6 CONCLUSION

Although our proposed method does not achieve any state-of-the-art (SOTA) results, it is not a black-box because it does not rely on any neural models. Furthermore, our method has only one critical hyperparameter $\sigma$, and it does not require a training process, making it easy to use. Additionally, our method still has room for improvement in terms of performance. For example, we can address the limitation discussed in Section 3, employ some tricks, or enhance the quality of the dataset, among other possibilities.

Besides, according to the auto regressive Hypothesis from Anonymous (2024b), our method also supports auto regressive MTS forecasting, but the inference efficiency is not good enough, we will not further discuss it in this paper.

Finally, the method proposed in this paper is strong evidence of Indeterminate Probability Theory. We hope that more people will join us in further developing this theory and exploring additional applications of Indeterminate Probability Theory.

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

## A   MATHEMATICAL FORECASTING PROOF OF PERIODIC CONTINUOUS FUNCTION

In this section, we rigorously prove that our proposed method can provide a strict and exact forecasting for a general periodic continuous function.

*Proof.* For a general periodic continuous function after transformation, we have $f(t) \in [0, 1], t \in \mathbb{R}$, and $f(t + m \cdot T) = f(t), m \in \mathbb{Z}, T$ is the period.

The continuous sequential indeterminate probability is formulated as

$$P(z \mid x_t) = \mathcal{N}(z; f(t), \sigma^2) \tag{15}$$

$$P(y_1 \mid x_t) = f(t) \tag{16}$$

with Eq.(9) we have an infinite joint indeterminate space as

$$P(z^{:-L} \mid x_t) = \prod_{\tau=0}^{L} P(z \mid x_{t-\tau}) = \prod_{\tau=0}^{L} \mathcal{N}(z; f(t-\tau), \sigma^2) \tag{17}$$

let $[a, b]$ be enough large interval of function $f(k)$, $b - a > L + D + T$ and the observations within this interval is for statistical calculation, we have

$$P(z^{:-L}) = \frac{\int_{k=L+a}^{b-D} P(z^{:-L} \mid x_k) dk}{b - a - D - L} \tag{18}$$

Substitute Eq.(18), Eq.(15) and Eq.(16) into Eq.(11), the forecasting problem is formulate as

$$P^{z^{:-L}}(y_1^{+d} \mid x_t) = \int_{z^{:-L}} \left( \frac{\int_{k=L+a}^{b-D} \left( P\left(y_1 \mid x_{k+d}\right) \cdot P\left(z^{:-L} \mid x_k\right)\right)}{\int_{k=L+a}^{b-D} P\left(z^{:-L} \mid x_k\right)} \cdot P\left(z^{:-L} \mid x_t\right) \right) \tag{19}$$

$$= \mathbb{E}_{z \sim P(z \mid x_{t-\tau})} \left[ \frac{\int_{k=L+a}^{b-D} \left( P\left(y_1 \mid x_{k+d}\right) \cdot \prod_{\tau=0}^{L} P\left(z \mid x_{k-\tau}\right)\right)}{\int_{k=L+a}^{b-D} \prod_{\tau=0}^{L} P\left(z \mid x_{k-\tau}\right)} \right] \tag{20}$$

$$= \mathbb{E}_{z \sim \mathcal{N}(z; f(t-\tau), \sigma^2)} \left[ \frac{\int_{k=L+a}^{b-D} \left( f(k+d) \cdot \prod_{\tau=0}^{L} \mathcal{N}(z; f(k-\tau), \sigma^2)\right)}{\int_{k=L+a}^{b-D} \prod_{\tau=0}^{L} \mathcal{N}(z; f(k-\tau), \sigma^2)} \right] \tag{21}$$

And

$$\lim_{\sigma \to 0} \prod_{\tau=0}^{L} \mathcal{N}(z; f(k-\tau), \sigma^2) = \begin{cases} \prod_{\tau=0}^{L} \mathcal{N}(z; f(t-\tau), \sigma^2), & \text{for } k = t + m \cdot T, \\ 0, & \text{for } k \neq t + m \cdot T, \end{cases} \quad (22)$$

where $z \sim \mathcal{N}(z; f(t-\tau), \sigma^2)$ and $L > T$.

Let $M = \lfloor \frac{b-a}{T} \rfloor$, and substitute Eq.(22) into Eq.(21)

$$P^{z^{:-L}}(y_1^{+d} \mid x_t) = \frac{M \cdot \left( f(t+d) \cdot \prod_{\tau=0}^{L} \mathcal{N}(z; f(t-\tau), \sigma^2) \right)}{M \cdot \prod_{\tau=0}^{L} \mathcal{N}(z; f(t-\tau), \sigma^2)}, \sigma \to 0. \quad (23)$$

$$= f(t+d) \quad (24)$$

$$= P(y_1^{+d} \mid x_t) \quad (25)$$

In this way, we have proved that our proposed method can make a strict and exact forecasting for periodic $f(t)$.

$\square$

## B    EXPERIMENTAL DETAILED SETTINGS

Table 4: Details of main experimental hyperparameter settings.

| | Metric | Latent Space | Past Length $L+1$ | Forget Number | Monte Carlo $C$ | Gaussian $\sigma$ |
|---|---|---|---|---|---|---|
| ETTh1 | 24 | 7*24-D | 24 | 10000 | 32 | 0.3 |
| | 48 | 7*48-D | 48 | 10000 | 16 | 0.4 |
| | 168 | 7*64-D | 64 | 10000 | 16 | 0.6 |
| | 336 | 7*64-D | 64 | 10000 | 16 | 0.6 |
| | 720 | 7*64-D | 64 | 10000 | 16 | 0.6 |
| ETTm1 | 24 | 7*3-D | 3 | 10000 | 32 | 0.1 |
| | 48 | 7*48-D | 48 | 10000 | 16 | 0.4 |
| | 96 | 7*64-D | 64 | 10000 | 16 | 0.6 |
| | 288 | 7*64-D | 64 | 10000 | 16 | 0.6 |
| | 672 | 7*64-D | 64 | 10000 | 16 | 0.6 |
| WTH | 24 | 21*3-D | 3 | 10000 | 8 | 0.4 |
| | 48 | 21*3-D | 3 | 10000 | 8 | 0.4 |
| | 168 | 21*3-D | 3 | 10000 | 8 | 0.4 |
| | 336 | 21*3-D | 3 | 10000 | 8 | 0.4 |
| | 720 | 21*3-D | 3 | 10000 | 8 | 0.4 |
| ECL | 48 | 321*4-D | 4 | 5000 | 8 | 1 |
| | 168 | 321*4-D | 4 | 5000 | 8 | 1 |
| | 336 | 321*4-D | 4 | 5000 | 8 | 1 |
| | 720 | 321*4-D | 4 | 5000 | 8 | 1 |
| | 960 | 321*4-D | 4 | 5000 | 8 | 1 |
| ILI | 24 | 7*90-D | 90 | 10000 | 64 | 1.4 |
| | 36 | 7*90-D | 90 | 10000 | 64 | 1.4 |
| | 48 | 7*90-D | 90 | 10000 | 64 | 1.4 |
| | 60 | 7*90-D | 90 | 10000 | 64 | 1.4 |
| Traffic | 24 | 862*2-D | 2 | 2000 | 8 | 1.8 |
| | 48 | 862*2-D | 2 | 2000 | 8 | 1.8 |
| | 168 | 862*2-D | 2 | 2000 | 8 | 1.8 |
| | 336 | 862*2-D | 2 | 2000 | 8 | 1.8 |
| | 720 | 862*2-D | 2 | 2000 | 8 | 1.8 |

- Forget Number is discussed in CIPNN.

- Monte Carlo $C$ is not critical, use smaller value for faster inference.

