# OpenReview forum: "Sequential Indeterminate Probability Theory for Multivariate Time Series Forecasting"
_ICLR.cc/2024/Conference — ICLR 2024 Conference Withdrawn Submission_

### Official Review · Reviewer_hhu6 · 2023-10-18

**Soundness:** 1 poor
**Presentation:** 1 poor
**Contribution:** 2 fair
**Rating:** 1
**Confidence:** 4

**Summary:**

The manuscript proposes a probabilistic model for multi-variate time series forecasting, which is not a black-box model.
The proposed method is based on the Indeterminate Probability Theory[1], modelling the conditional distribution of the predictions given the context, via some latent random variables.
The results show that the proposed model does not achieve any SOTA results, but it is claimed that the method outperforms LSTM models as well as some transformer-based models.


[1]Anonymous. Indeterminate probability theory. ICLR 2024 Submission ID 4295, Supplied as additional
material., 2024b.

**Strengths:**

The manuscript aims at tackling the multi-variate time series forecasting task using a non-black-box model.
Originality: The attempt of employing a model that "does not require a training process" could give new insights to the time series modelling community.
The quality and clarity still have space to improve.
The manuscript claims that the method outperforms LSTM models as well as some transformer-based models.

**Weaknesses:**

The manuscript proposes a model for multi-variate time series forecasting.
However, the manuscript is not well self-contained.
For instance, the background and the setup of the forecasting task is not well demonstrated;
the theoretical contribution is highly dependent on [1], but I failed to figure out the key contribution of the manuscript upon [1].

The language could be improved.

The authors might want to distinguish the use of ```\citep{}``` and ```\citet{}```, and update the references with their latest status, instead of the arXiv version.

LSTM is mentioned in the abstract and the experimental section but not discussed in related works.

Figure 4 is not discussed in the manuscript.

From my humble opinion, the manuscript needs a revision before publishing.

**Questions:**

I could not find the connection between eq (1) and [1]Anonymous (2024b) clearly, can you elaborate how eq (1) is inferred from it?

What does "with duplicated random variables" mean in the abuse test?

---

### Official Review · Reviewer_xZ9w · 2023-10-25

**Soundness:** 1 poor
**Presentation:** 1 poor
**Contribution:** 1 poor
**Rating:** 1
**Confidence:** 4

**Summary:**

This paper builds on a method called Indeterminate Probability Theory that allegedly solves the problem of deriving a general posterior. By applying the theory to multivariate time series (MTS) forecasting, the paper proposes a probabilistic method using Monte Carlo for estimating conditional distributions.

**Strengths:**

- the benchmark seems to have been fairly evaluated

**Weaknesses:**

- Writing could benefit from further editing, including
    + sentences that are too short to understand them (especially in the introduction / notation, see below)
    + sentences that are grammatically not completely correct
    + typos
    + etc.
- Technically not sound (and so are the two references (Anonymous, 2023a,b) articles the paper is based on)
    + The described setup is unclear,
        * e.g., what exactly is $\textbf{z}$?
        * what is meant with $P(x_k)=1$? The experiments condition on $x_k$?
        * what is meant with the $k$th experiment?
        * what do you mean with "we can infer $Y = y_l, ...$"?
        * with respect to what measure is integrated in (1)?
        * what exactly is $P^Z$, because in (1) Z is integrated out and hence the posterior should not depend on Z
        * etc.
    + There are implicit assumptions made in the paper that are not discussed nor do they seem to be realistic:
        * $P(\textbf{z} | x_k)$ can be factorized (implicitly assumed in (2)) / $\textbf{z}$ are indepdent. Later $N$ is used for the number of time series, which would imply independence of time series -- and hence no need for an MTS approach
        * Eq. 5 and 6 implicitly assumes strong stationarity (if I understand their notation correctly -- not entirely clear though)
     + if the method does not require training, how do you obtain $\sigma^2$ in (14)?
     + etc.

**Questions:**

Maybe the authors can clarify some of the above mentioned weaknesses.

---

### Official Review · Reviewer_6gm7 · 2023-11-01

**Soundness:** 2 fair
**Presentation:** 1 poor
**Contribution:** 2 fair
**Rating:** 1
**Confidence:** 3

**Summary:**

This paper applies a concurrently-submitted approach denoted "Indeterminate Probability Theory" to the problem of multivariate timeseries forecasting. While none of the results are state of the art, the authors posit that reasonable performance on an assortment of datasets validates the approach, and that this approach provides a non-blackbox approach to MTS forecasting.

**Strengths:**

*Originality*

Being concurrently submitted, "Indeterminate Probability Theory" is here newly applied to multivariate timeseries forecasting.

*Quality*

Experimental results are given which apply the approach to multiple multivariate timeseries datasets.

*Clarity*

Needs work.


*Significance*

The paper presents an early application of "Indeterminate Probability Theory". The lack of need for a training process is a potentially interesting contribution.

**Weaknesses:**

*Originality*

The application domain is familiar to the community. Most of the novelty of this work rests on its use of "Indeterminate Probability Theory". Arguably, both of the application papers ought to be merged with the theory proposal, to provide demonstrations of its applicability. Such a work might become too long, and perhaps more appropriate to a journal submission.

*Quality*, *Clarity*

The quality of prose and exposition in this submission is poor. Indeterminate Proability Theory is inadequately explained in this submission, forcing the reader to the supplementary material for the concurrent submission. Even there, it is difficult to take at face value such broad claims such as "we only need sample two points (C = 2), even for a 1000-dimensional latent space".
The authors attribute poor results on WTH dataset to an explanation in Section 3 without further comment, but I cannot find such an explanation.


*Significance*

A lack of parameters to optimize or even hyperparameters to tune seems likely to limit the applicability of the approach.

**Questions:**

Univariate Informer achieves substantially better results than multivariate. Why should that not be the baseline model cited in table 2? Even a univariate ARIMA model achieves 0.396 0.504 MSE/MAE on ETTh1@168. Univariate models are arguably simpler, more easily explained, and achieving better metrics, so it's surprising to see them not compared.

Continuing with ETTh1, "Are Transformers Effective for Time Series Forecasting?" NLinear approach achieves even better metrics than neural approaches, yet does not appear in Table 2.

---

### Official Review · Reviewer_TTQE · 2023-11-06

**Soundness:** 2 fair
**Presentation:** 1 poor
**Contribution:** 2 fair
**Rating:** 3
**Confidence:** 2

**Summary:**

This paper proposes a sequential indeterminate probability theory approach for multivariate time series forecasting. Indeterminate probability theory is a new approach proposed as a part of another ICLR submission, and has been referred to in this paper. The paper proposes a closed form solution to the future predictions conditioned on the past predictions. Results on benchmark datasets are worse than state of the art approaches.

**Strengths:**

- The paper proposes an interesting approach for multi-variate time series forecasting using indeterminate probability theory and predicts the future time series as a closed form.
- The proposed closed form is not a blackbox and can be used to explain the prediction.

**Weaknesses:**

- The paper builds upon another work submitted at the same conference. The core ideas of indeterminate probability theory have not been presented in the paper, making this paper hard to read. The details  about the implementation of the prediction function have also been omitted.
- The experimental performance of the proposed method is much worse than the baselines, as such there is a limited utility for this approach.
- Simple baselines such as linear models are missing from the results table.

**Questions:**

Equation 14 provides the exact closed form solution for the future predictions. Could the authors elaborate further on this equation? What is the high level intuition of such an approach working in practice?